# Rapid Eocene diversification of spiny plants in subtropical woodlands of central Tibet

Xinwen Zhang [1,2,12], Uriel Gélin[3,12], Robert A. Spicer [1,4], Feixiang Wu[5,6], Alexander Farnsworth [7,8], Peirong Chen[1,2], Cédric Del Rio [1,9], Shufeng Li[1,2,10], Jia Liu[1,10], Jian Huang[1,10], Teresa E. V. Spicer[1], Kyle W. Tomlinson [3], Paul J. Valdes [7], Xiaoting Xu[1,2], Shitao Zhang[11], Tao Deng [5,6], Zhekun Zhou [1,10] & Tao Su [1,2,10✉]

Spinescence is an important functional trait possessed by many plant species for physical defence against mammalian herbivores. The development of spinescence must have been closely associated with both biotic and abiotic factors in the geological past, but knowledge of spinescence evolution suffers from a dearth of fossil records, with most studies focusing on spatial patterns and spinescence-herbivore interactions in modern ecosystems. Numerous well-preserved Eocene (~39 Ma) plant fossils exhibiting seven different spine morphologies discovered recently in the central Tibetan Plateau, combined with molecular phylogenetic character reconstruction, point not only to the presence of a diversity of spiny plants in Eocene central Tibet but a rapid diversification of spiny plants in Eurasia around that time. These spiny plants occupied an open woodland landscape, indicated by numerous mega-fossils and grass phytoliths found in the same deposits, as well as numerical climate and vegetation modelling. Our study shows that regional aridification and expansion of herbivorous mammals may have driven the diversification of functional spinescence in central Tibetan woodlands, ~24 million years earlier than similar transformations in Africa.

[1] CAS Key Laboratory of Tropical Forest Ecology, Xishuangbanna Tropical Botanical Garden, Chinese Academy of Sciences, Mengla, China. [2] University of Chinese Academy of Sciences, Beijing, China. [3] Center for Integrative Conservation, Xishuangbanna Tropical Botanical Garden, Chinese Academy of Sciences, Mengla, China. [4] School of Environment, Earth and Ecosystem Sciences, The Open University, Milton Keynes, UK. [5] CAS Key Laboratory of Vertebrate Evolution and Human Origins, Institute of Vertebrate Paleontology and Paleoanthropology, Chinese Academy of Sciences, Beijing, China. [6] CAS Center for Excellence in Life and Paleoenvironment, Chinese Academy of Sciences, Beijing, China. [7] School of Geographical Sciences and Cabot Institute, University of Bristol, Bristol, UK. [8] State Key Laboratory of Tibetan Plateau Earth System, Environment and Resources, Institute of Tibetan Plateau Research, Chinese Academy of Sciences, Beijing, China. [9] CR2P - Centre de Recherche en Paléontologie – Paris, MNHN - Sorbonne Université - CNRS, Paris, France. [10] Center of Plant Ecology, Core Botanical Gardens, Chinese Academy of Sciences, Mengla, China. [11] Faculty of Land Resource Engineering, Kunming University of Science and Technology, Kunming, China. [12] These authors contributed equally: Xinwen Zhang, Uriel Gélin. ✉email: sutao@xtbg.org.cn

Plant functional traits refer to plant characters that impact plant survival, growth, or reproduction[1,2], which in turn have significant influences on ecosystem processes[3,4]. Functional traits enable plants to better acquire and retain resources, promote niche differentiation, and reduce interspecific competition, while functional trait redundancy maintains the stability of ecosystem function[5–7]. Although most studies focus on the role of functional traits in structuring modern ecosystems[8–10], little is known about how these functional traits evolved in the geological past, particularly under varied environmental stressors. Plant functional traits are shaped by the interaction between plants and the surrounding environment, including faunal interactions, during the long process of evolution in deep time[1]; therefore, it is crucial to investigate the evolutionary history of plant functional traits to better understand the mechanisms moulding them.

Spinescence (a general term for the phenomenon of spines, prickles, and thorns on plants) is an important functional trait shared by numerous plant families worldwide and mainly provides physical protection against vertebrate herbivores[11,12]. Sharp spines can hurt the bodies and mouthparts of herbivores, and thus restrict their feeding rates[13]. In general, spine architecture and spatial distribution are largely influenced by mammal herbivory and correlate strongly with the spatial distribution of megaherbivores[13–16]. Both prickles and thorns can defend against herbivores and even climbing mammals[17], but whether they have other functional properties is unclear. Today, spinescence is common in open habitats with abundant herbivores and relatively arid climates, such as savanna, featuring an open tree canopy with a grassy understory[18–20].

To better understand mechanisms underlying the evolution of plant spines, it is necessary to explore their occurrence in deep time. Spiny plants in Africa, evidenced by both plant and mammal phylogenies, underwent a massive radiation during the early Miocene, particularly in African savannas[18], which are characterised as open-canopied ecosystems. Spiny species have been linked to the arrival and diversification of bovids from Eurasia during the Neogene, as well as climatic drying in Africa that promoted the development of open vegetation[18,21]. Even though spiny plants are distributed worldwide, our understanding of their evolutionary history remains woefully incomplete.

Fossils of plant spines are physical evidence for the presence of plant spinescence at the place and time the plant was alive; nevertheless, they have been largely ignored and scarcely reported compared to other plant organs, such as leaves, fruits, and seeds. Previously, spiny fossils have been largely overlooked in fossil floras, with a few exceptions such as the Eocene Green River flora[22] and Oligocene Bridge Creek flora[23,24] of North America, and the Miocene Tortonian flora[25] of Europe. Overall, their occurrence in fossil floras is not well documented and the ecological and evolutionary significance of these scattered spiny fossils have never been explored at continental scale.

Here we report exceptionally rich assemblages of spiny plant fossils collected from late Eocene (~39 Ma) sediments in central Tibet (Fig. 1). The Tibetan region has undergone dramatic evolutionary and climatic change since the collision of the Indian and Eurasian plates began between 65–55 Ma[26–30]. Habitat differentiation favoured the diversification and turnover of plant and mammal species[31,32] as central Tibet changed from hosting a closed subtropical humid lowland forest valley ecosystem during the middle Eocene[33] to the open and dry highland steppe of today[34]. These spiny fossils together with phytoliths and plant megafossils, including those of monocotyledonous herbs and dicotyledonous woody species, point to an initial opening-up of the vegetation as early as the late Eocene in central Tibet. Combined with molecular phylogenetic analyses, we document the early diversification history of spiny species in Eurasia. Using proxy and modelling data, we reconstruct the vegetation, climate and herbivory that favoured spiny plant evolution in the Eocene of central Tibet.

## Results

**Geological age**. We document a total of 44 spine-bearing fossil specimens collected from two fossil localities (Fig. 1, Supplementary Fig. 1): the Dayu locality (32° 20′ N, 89° 46′ E), within the middle member of the Niubao Formation, is considered to be ~39 Ma (Bartonian, early late Eocene) based on radiometric (U/Pb) dating[35,36], while the other site at Xiede (31° 58′ N, 88° 25′ E) we consider age-equivalent as it contains a wide range of similar fossils in addition to spine-bearing plants. Both localities are within the central Tibetan Bangong-Nujiang Suture Zone.

**Spine morphology**. Spine morphology is here divided into prickles (modified epidermis) and thorns (modified axial stems). According to size and growth pattern, we classify prickles into two types and thorns into five types. Further details are given in Supplementary Figs. 2–8, Supplementary Note 1.

*Prickles*. Morphotype I (Fig. 2b) Prickles arranged on stems alternately. The average length of prickles is 4.9 ± 1.6 mm. The mean width at the base of each prickle is 5.8 ± 1.2 mm. Each prickle curves upward at an angle between 45° and 90°.

Morphotype II (Fig. 2e) Prickles occur on stems irregularly. The average length of prickles is 3.8 ± 1.2 mm. The mean width at base of each prickle is 5.2 ± 1.5 mm. The prickles grow almost perpendicular to the stem.

*Thorns*. Morphotype III (Fig. 2h) Thorns grow on stems oppositely and densely. The average length of thorns is 5.6 ± 2.1 mm. They grow on the stem at an angle of ~90°.

Morphotype IV (Fig. 2i) Thorns grow on stems alternately. The average length of thorns is 4.7 ± 1.7 mm. Each thorn curves slightly upwards or downwards.

Morphotype V (Fig. 2d) Thorns grow on stems alternately. The average length of thorns is 29.0 ± 14.7 mm. They grow on the stem at an angle of ~90°.

Morphotype VI (Fig. 2j) Thorns grow on stems alternately and at an acute angle of less than 45°. The average length of thorns is 5.9 ± 1.4 mm.

Morphotype VII (Fig. 2k) Thorns grow on stems in pairs oppositely. The average length of thorns is 2.6 ± 0.6 mm. They grow on the stem at an angle of ~90°.

**Phylogeny of spiny plants**. The accumulation curve of spiny eudicots (Fig. 3a) suggests that the first spiny plant species emerged during the Paleogene in Eurasia. This strengthens our findings pointing to an early occurrence of spiny plants in Asia, although fossil spines are generally under-explored over this period. The proportion of spiny plants remained low until 40 Ma, but has risen exponentially since then (Fig. 3a). For approximately 20 Myr after their emergence we estimate that spines occurred in only 2 lineages, but during the late Eocene there was a four-fold increase from 2 to 8 spiny lineages within the next 10 Myr, closely matching the 7 morphotypes found in our fossil records. The beginning of this substantial radiation of spiny plants, estimated according to molecular clock techniques, closely matches the timing of our fossil spines, reinforcing the hypothesised early emergence of spiny plant diversity in Asia. The mid-late Eocene, therefore, witnessed a step change in the abiotic and/or biotic environment of the woody flowering plants in Eurasia.

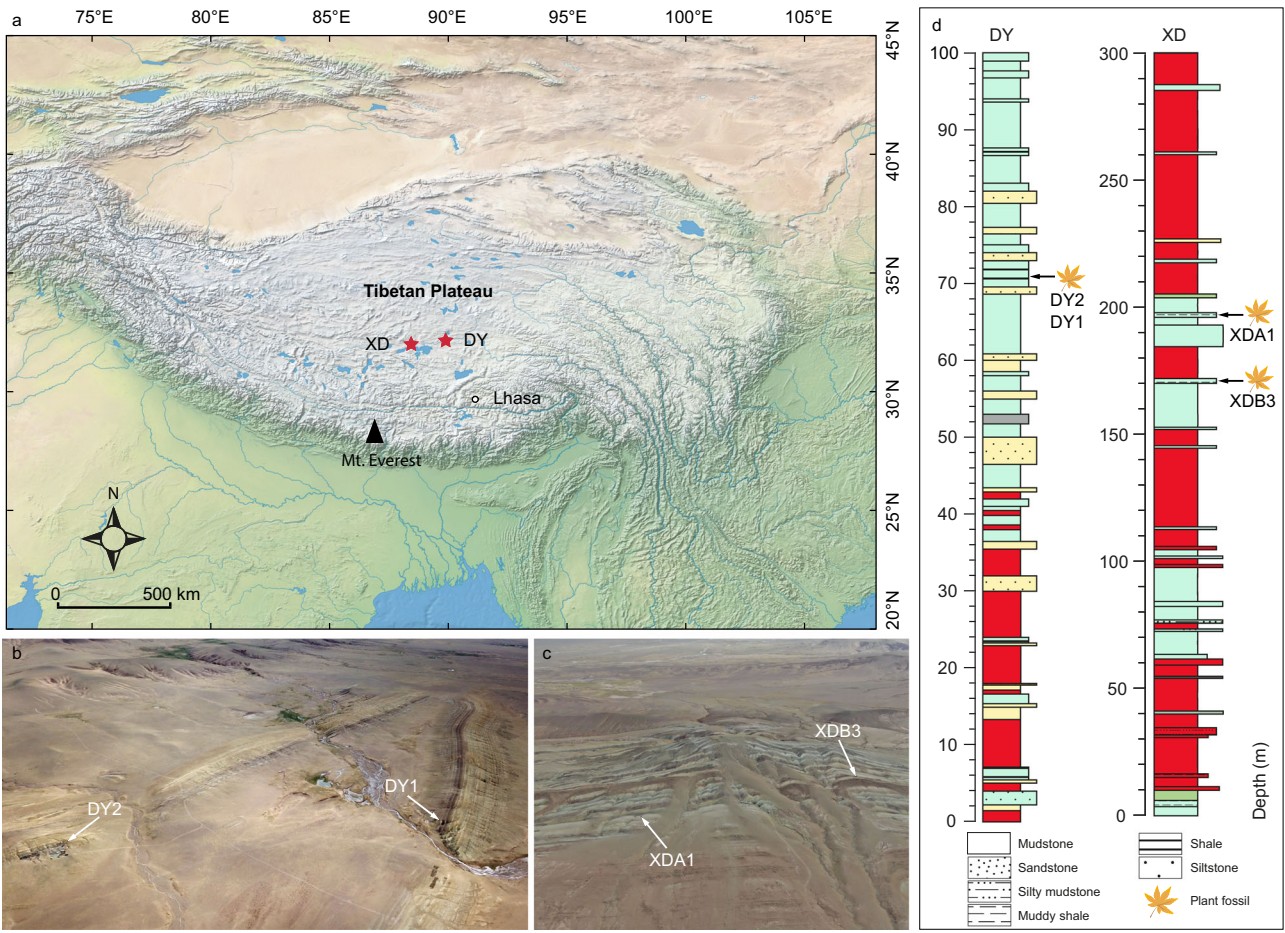

**Fig. 1 The location of two fossil sites from the early late Eocene of central Tibetan Plateau, China. a** Map showing the fossil sites (red stars) from the middle member of the Niubao Formation, central Tibetan Plateau. The base map was downloaded from Natural Earth (https://www.naturalearthdata.com/). XD, the site in Xiede village; DY, the site in Dayu village. **b**, **c** Outcrop drone images of fossil sites near Dayu[51] and Xiede villages. **d** The stratigraphy of Dayu[51] and Xiede sections. The colour of each layer reflects the colour of the rock.

**Palaeoclimate and palaeoelevation.** We applied the Climate-Leaf Analysis Multivariate Program (CLAMP) to the Dayu leaf flora (Supplementary Figures 9, 10), which indicates a climate with summers that were warm (warm month mean temperatures of 24–30 °C) and dry (summer vapour pressure deficit (VPD.sum) of 10.2–17.2 hPa), and winters that were cool (cold month mean 0.2–7.2 °C) and moist (VPD.win 2.8–5.8 hPa). Occasional frosts may have occurred but were not harsh, nor prolonged, and thus survivable by palms that also occur in the Dayu section. The dry bulb mean annual temperature of 15.6 °C exceeds, but is close to, the lower survivable limit for palms (14.2 °C)[37]. The wet/dry precipitation ratio is 4.6:1, suggesting a borderline monsoonal climate. More details are given in Supplementary Tables 1–3.

Both the moist enthalpy and the wet bulb terrestrial lapse rate approaches for palaeoaltimetry[38] gave similar results for the elevation of the Dayu fossil assemblage: 2.6 ± 1.2 km and 2.7 ± 0.9 km respectively (Supplementary Note 2).

Numerical climate modelling for central Tibet, with a valley floor set at 2.5 km bounded by 5 km high East-West trending mountain ranges, also indicates a winter-wet warm climate within the valley (a mean annual air surface temperature 23–26 °C, and a cold month mean surface air temperature of 9–11 °C depending on location) with progressively more arid conditions towards the east where some minor summer rain also is predicted (Supplementary Note 3). Note that these model temperatures are slightly warmer than those reconstructed from CLAMP, but the nominal model valley floor elevation is lower than that

reconstructed from proxies, and CLAMP is known to return dry bulb temperatures that reflect evapotranspirational cooling in dry regimes where groundwater is plentiful[39]. Overall, the modelling and proxy thermal regimes are similar, pointing out to drying and cooling climate in central Tibet by the mid-late Eocene accompanied by some within-valley increase in elevation (~1 km) since ~47 Ma[33].

**Herbaceous plant diversity.** Herbaceous fossil specimens of the Dayu flora ($n = 315$), classified as monocots based on their parallel veins and gross morphology, account for ~38% of all plant fossil specimens found at Dayu. Using leaf size, length, and stem growth form, we were able to classify herbaceous fossils into six morphotypes (Supplementary Fig. 11). Currently, we cannot determine if they were terrestrial or aquatic plants due to limited preservation. However, there are abundant herbaceous phytoliths in the bounding sediments, representing several mainly non-aquatic grass subfamilies, including Chloridoideae and Pooideae (Supplementary Figure 12). The diverse morphology of phytoliths observed from the fossil-bearing outcrops (Supplementary Figure 12) indicates a species-rich grass community within an ecosystem that also contained palms and numerous other woody taxa, but evidence for such a diverse grass component does not occur in older Tibetan floras, e.g., the middle Eocene Jianglang flora from central Tibet[33] (Supplementary Fig. 13).

Progression from the older flora to the younger in the Bangor and Lunpola basins shows an increase in phytolith abundance

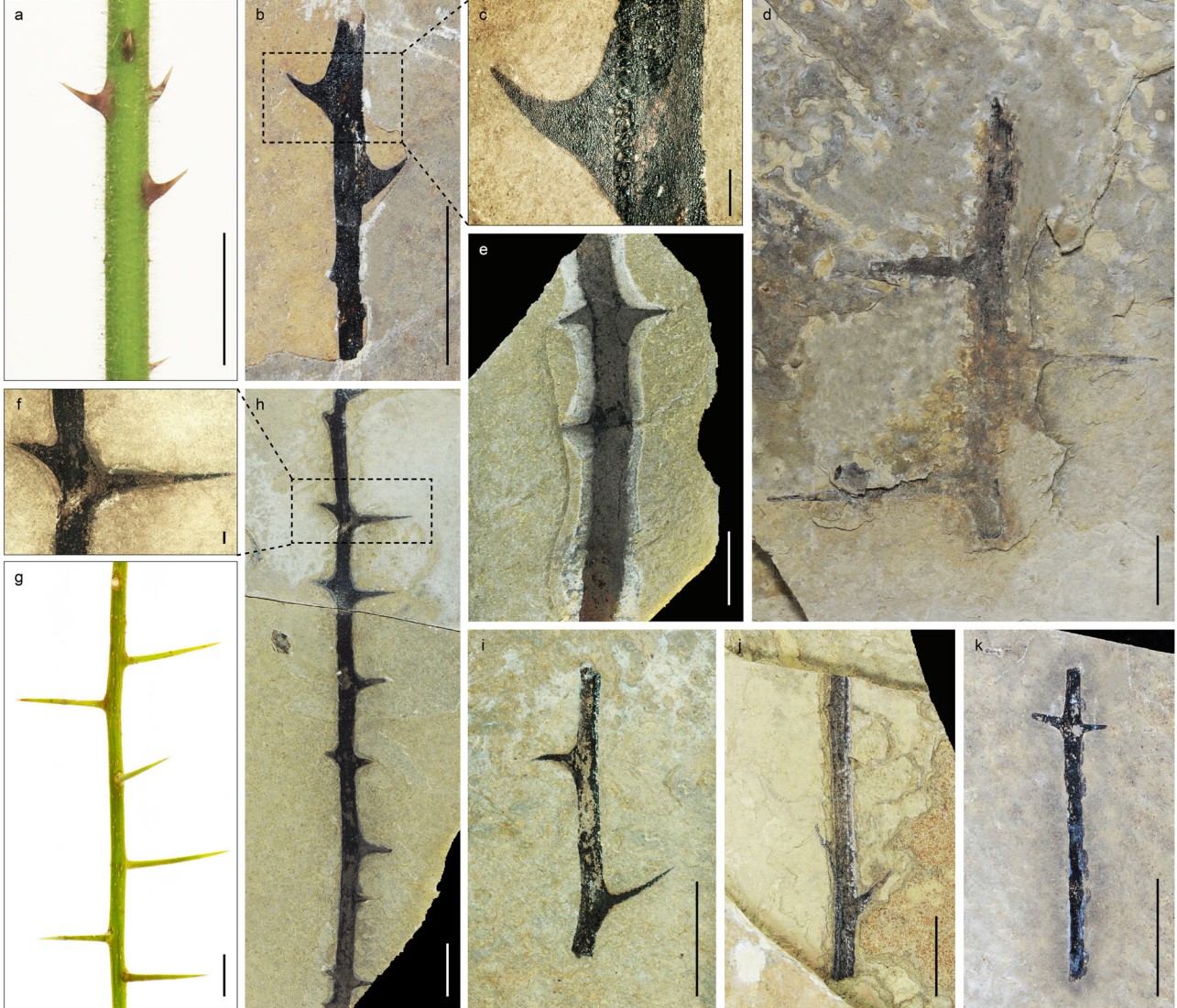

**Fig. 2 Morphotypes of spiny fossils from the upper Eocene Dayu and Xiede sections, central Tibetan Plateau. a** Prickles of living species *Rubus alceifolius*. **b, d, e, h–k,** Seven morphotypes of spiny fossils. **c** Enlargement of **b. f** Enlargement of **h. g** Thorns of living species *Xylosma racemosum*. (For **c, f,** scale bars = 1 mm; for others, scale bars = 10 mm).

and diversity. The phytolith assemblage from the ~47 Ma Jianglang section, where the megafossils are interpreted to represent a subtropical forest, is dominated by those forms produced by woody taxa, and these greatly exceed grass forms (Supplementary Fig. 13, Supplementary Table 4). Within the lower part of the Dayu section (~39 Ma), phytoliths are still dominated by those produced by woody plants but bulliform phytoliths typically produced by grasses become more common (Supplementary Figure 13). The main fossil-bearing layer higher in the Dayu section preserves not only the numerous spiny taxa but also abundant phytoliths where, among those that could be identified, 66% were produced by grasses and only 34% by woody plants. As grass tends not to grow in abundance in shaded conditions, all these megafossils and phytoliths strongly suggest the progressive development of a semi-open habitat dominated by herbaceous plants with trees forming a broken canopy, such as is seen in modern woodlands[40,41]. This interpretation is further supported by experiments using fully coupled ocean-atmosphere-vegetation climate modelling with Eocene boundary conditions and a Tibetan Central Valley topography (Supplementary Note 3, Supplementary Fig. 14). These simulations show early Eocene closed forest vegetation transitioning to a more open system by the middle Eocene.

**Herbivorous mammal fossils of the Tibetan Plateau**. Available fossil records indicate that the regional diversity of large herbivorous mammals increased substantially from the Paleogene to the Neogene (Fig. 3a). Large herbivores were abundant in central Asia after the early Eocene. There are numerous fossil records of Paleogene herbivorous mammals from the Tibetan Plateau and its surrounding regions, mainly represented by Brontotheriidae, Hyracodontidae, Amynodontidae, Paraceratheriidae, and Anthracotheriidae (Supplementary Data 1), and recently a rhino (*Plesiaceratherium* sp., Rhinocerotidae) was found from the early Miocene strata of the central Tibetan Plateau[42]. At the broader scale of central Asian countries, we observed a first peak of herbivore diversification in the first half of the Eocene (~50 Ma, Fig. 3a), followed by a second peak between the late Eocene to early Oligocene (~40–28 Ma; Fig. 3a). Most of the second peak species belonged to Perissodactyls (~84%, *n* = 187), mainly from the Hyracodontidae, Brontotheriidae and Lophialetidae families. While most of the species identified to family level in the late

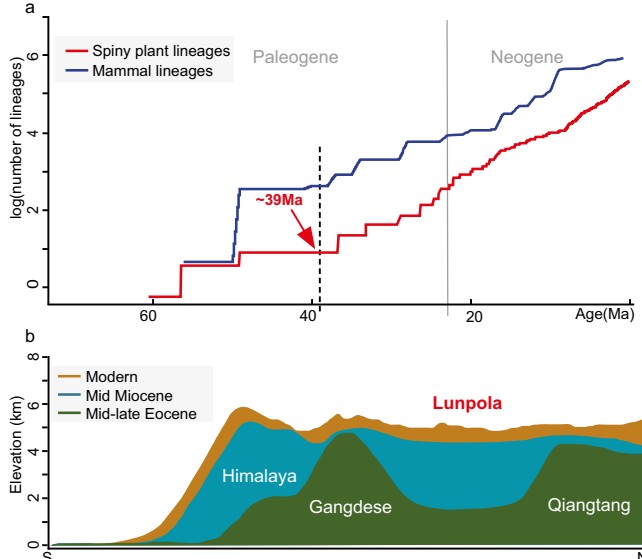

**Fig. 3 The lineage accumulation curve of spiny plants in Asia and the elevational changes of central Tibetan Plateau. a** The red line represents a log plot of lineage accumulation of spiny plants in eudicots in Eurasia during the Cenozoic. The blue curve represents a log plot of lineage accumulation of mammalian herbivore species in central Asia during the Cenozoic. Being log plots, the linear rise is indicative of exponential diversification. Source data are provided as Source Data files. **b** South (left) to north (right) transects of mean elevation changes across Himalaya-Tibet at different phases in its geological evolution (modified from Su, et al.[51]).

Eocene still belonged to Brontotheriidae, a few species of Hyracodontidae and Lophialetidae emerged, suggesting that these clades may have helped drive the spiny plant diversification reflected by our findings. Among the main changes in the late Eocene-early Oligocene fauna, the proportion of Artiodactyls increased to ~43% (n = 124), Rhinocerotidae, Paraceratheriidae and Lophiomerycidae started to diversify, and finally cervids and bovids appeared (Supplementary Data 1). In addition, many hippo-like amphibious species belonging to Anthracotheriidae, Amynodontidae and Anthracobunidae were present both in the first half of the Eocene and in the late Eocene–early Oligocene, with 30 and 36 species respectively. This is consistent with sedimentological evidence for the presence of water bodies, at least seasonally, which over time became increasingly brackish to saline[43]. Furthermore, these clades included several very large herbivores that required large amounts of food, both grass and tree leaves. These megaherbivores have been found since in open habitats such as savanna[44], adding to our interpretation that increasing herbaceous (predominantly grass) and spiny plants coexisted in an open canopy semi-wooded environment.

## Discussion

### The late Eocene diversification of spiny plants and semi-open woodlands in central Tibet.
The rich assemblage of spiny plant fossils from the late Eocene (~39 Ma) along the Bangong-Nujiang Suture Zone in central Tibet traces the evolutionary history of spiny plants in Eurasia back to the Paleogene. These fossils evidence an early diversification of spiny plants in the Tibetan region contemporaneous with the emergence of open semi-wooded habitats by the late Eocene, and early in the transition of central Tibet to full plateau formation which appears to have been almost complete early in the Oligocene[36].

Based on current knowledge, this late Eocene flora in central Tibet bears the richest diversity of spiny plants known among

Cenozoic floras worldwide. The evolutionary history of spiny plants is still poorly known largely due to a dearth of fossil records, which may be because spinescence tends to be most prevalent in semi-arid to arid environments where fossilisation potential is normally low[18,45]. The rich spine assemblages from Tibet enable us to investigate the morphological diversity of spiny plants in a highly unusual context. The fossils studied here represent both prickles and thorns divisible into seven morphotypes (Fig. 2). Although with only morphological characters it is difficult to assign unambiguously these fossils to specific taxa, these spiny fossils show a range of distinct morphological characters, indicating that a wide variety of spiny species existed in the same community in central Tibet during the late Eocene.

All plant fossil evidence, together with numerical modelling (Supplementary Note 3; Supplementary Fig. 14), indicate that central Tibet hosted a landscape that supported an open habitat, seemingly in the form of woodlands, during the late Eocene. Woodlands are usually considered to be tree-rich communities with open canopies and grassy understories[40,41]. Today, spiny plants appear to be most abundant in open canopy communities, where predation pressure from mammalian herbivores is high[11,18]. Numerous fossilised monocots (Fig. 4, Supplementary Fig. 11) and abundant phytoliths (Fig. 4, Supplementary Fig. 12) representing the grass family (Poaceae) and attributable to the subfamilies Chloridoideae, Pooideae, and potentially Bambusoideae point to an open ecosystem[46]. These well-preserved fossils of herbaceous plants show that the vegetation could not have been dense forest, even though the assemblage inevitably will have been biased towards representing a more tree-rich community bordering the ancient lake shoreline where water was most abundant. This is emphasised by the presence in the region of several amphibious mammals that had a similar niche to modern hippos[47] in African savanna. Away from lake margins, the relatively seasonally dry environment would support fewer trees and shrubs and thus more open vegetation. Woody species that have been found preserved in the Dayu section include *Koelreuteria lunpolaensis*[48], *Ailanthus maximus*[49], *Cedrelospermum tibeticum*[50], and several species in Malvaceae and Fabaceae (Supplementary Figs. 9, 10), which are families commonly found in open ecosystems, as well as a palm, *Sabalites tibetensis*[51]. Our census of the leaf flora from the same layer at the Dayu site shows that the modern affinities of 10 fossil taxa in the flora belong to families/orders that contain spine-bearing species, i.e., Malvaceae, Rosales, Fabaceae, Ulmaceae, Cannabaceae, Menispermaceae, Simaroubaceae, Anacardiaceae, Myrtaceae, and Araliaceae (Supplementary Table 5). Notably, the plant diversity was much lower than in the earlier middle Eocene (~47 Ma) Jianglang subtropical flora reported recently from the adjacent Bangor (Baingoin) Basin[33].

### Factors promoting the diversification of spiny plants in central Tibet.
The Tibetan Plateau experienced a complex geological evolution, leading to dramatic palaeoenvironmental changes both locally and regionally[52–54]. Although the details of Tibetan orogeny are still being resolved, the latest geological and palaeontological studies have revealed that the plateau did not rise as a single coherent entity[55–57]. A deep East-West trending valley existed in what is now the central Tibetan Plateau for much more of the Paleogene (Fig. 3b), with the valley bottom being at an elevation of ~1500 m at 47 Ma[33] and ~2600–2700 m at ~39 Ma (Supplementary Note 2), bounded by the Gangdese and the Tanggula (Qiangtang) highlands with crest heights > 4 km to the south and north[58] respectively. It was not until just prior to the Neogene that the modern low relief high elevation plateau formed[36,51,59]. During the middle Eocene, a warm and humid

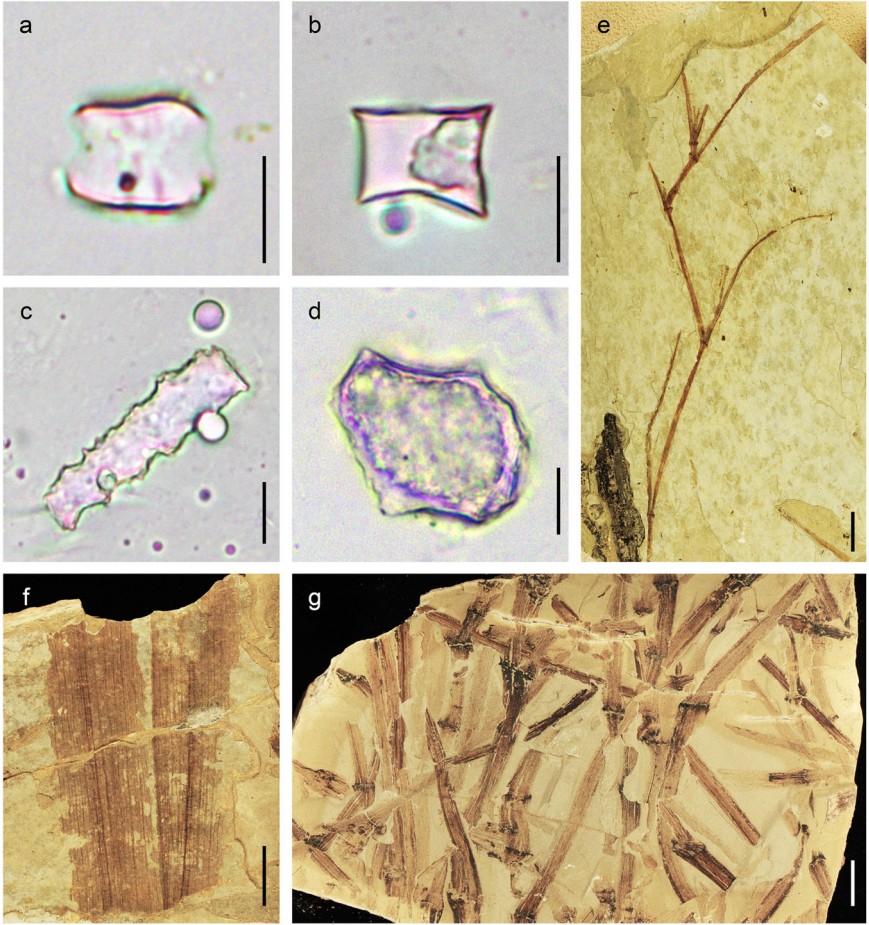

**Fig. 4 Typical herbaceous fossils and phytoliths from the Dayu section. a–d** Phytoliths in different forms extracted from the Dayu sediments. **a** True saddle; **b** Rondel; **c** Elongate; **d** Bulliform. **e**, **f** Herbaceous megafossils from Dayu section. **e** Sympodial branching; **f** Wide leaf with parallel veins; **g** Stem with swollen nodes. For phytolith analysis, we performed three replicate experiments with consistent results. (For **a–d**, scale bars = 10 μm; for **e**, **f**, scale bars = 10 mm).

airflow evidently penetrated the valley, supporting a humid climate and subtropical biota[33,54,60]. The palaeoclimate of the Dayu flora derived from the CLAMP analysis is similar, but slightly drier, than that experienced by the older Jianglang flora nearby (Supplementary Table 1), and is consistent with an early onset of regional drying that became more pronounced after the end of the Oligocene[43]. The reduction in growing season precipitation is largely a function of the shorter growing season, but the mean monthly growing season precipitation does indicate slightly drier conditions for the Dayu assemblage.

Undoubtedly, palaeoenvironmental changes during the development of the Tibetan Plateau stimulated an overturn of plant taxa and an opening-up of the vegetation as the floor of the central valley rose in the latter part of the Paleogene and regional drying took place. These changes may have stimulated herbivore access, increased herbivory pressure on plants, and drove the evolution of defensive spines. Moreover, at this time global climate began its transition from a 'warm house' to a 'cool house' and eventually an icehouse condition[61]. Nevertheless, the presence of palm fossils in the Dayu flora indicated that the coldest month mean temperature and mean annual temperature in central Tibet at ~39 Ma were not below 5.2 °C and 14.2 °C, respectively[51]. Temperature, precipitation and solar radiation are major abiotic factors that determine vegetation growth, distribution ranges and dynamic change[62]. When thermal conditions are sufficient for vegetation growth, water becomes the main limiting factor controlling vegetation status[63]. During the early late

Eocene, central Tibet was an intermontane lowland bounded by mountains that seem to have exceeded 4,000 metres[59]. Despite the initial influx of moisture, the retreat of the Neotethys during that period resulted in a reduction in the amount of water vapour entering the then shallowing valley, which, together with a growing Himalaya, led to the increasingly dry Neogene environmental conditions across the Tibet and central Asia[64]. A recent study demonstrates both drying and cooling occurred in central Tibet as the surface elevation of the Tibetan Central Valley increased from ~2.5 km to more than 4 km between 39 Ma and 29 Ma[36].

Since open habitats are generally able to carry larger groups of herbivores and these animals have a greater impact on plants whose growth is also limited by reduced rainfall and temperature, herbivore pressure during this period inevitably increased, driving the evolution of defensive spines. Numerous fossils of large herbivorous mammals have been found on the Tibetan Plateau and adjacent regions, evidencing their presence in the Paleogene (Supplementary Data 1). Examples include the late Eocene Brontotheriidae[65], Hyracodontidae, late Oligocene Paraceratheriidae[66], as well as the early Miocene Rhinocerotidae[42]. Although many species are now extinct, fossil records show that herbivorous mammals became increasingly diversified, including an important turnover of herbivore families, notably the emergence of bovids and cervids, from the Paleogene to the Neogene (Fig. 3a, the blue curve).

In terrestrial ecosystems, mammalian herbivores influence the length, density, and distribution of plant spines[14,67,68]. For example, the rapid accumulation of spiny plants in Africa since the middle Miocene closely matched the arrival of bovids[18]. The earlier emergence of ruminants, and other extant families of herbivores in Eurasia may have triggered the early radiation of spiny plants in Tibet by filtering out the undefended plant species. Mammalian herbivory is a major driver of global vegetation dynamics[69], and large animals are thought to have the capacity to create open ecosystems by reducing woody biomass[70-72]. Exclosure experiments conducted in Africa reveal that species-rich megaherbivore communities can effectively reduce woody vegetation coverage by 15–95%[73].

The plant fossils reported here record the beginnings of vegetation opening-up under a drying/cooling climate as the modern Tibetan Plateau began to form from the late Eocene onwards, and the observed spinescence marks the early development of defence mechanisms against large herbivore feeding pressure in the region. We infer that in the late Eocene climate changes in central Tibet, driven by the onset of global cooling[61] and regional tectonism[74], resulted in changes in vegetation that allowed increased access by large herbivores. This in turn led to further opening-up of the landscape favouring further diversification of herbivorous mammals and spiny plants. This string of climate/plant/animal interactions long pre-dated the middle Miocene arrival of herbivorous mammals (especially bovids) in Africa where similarly linked evolutionary feedback processes took place some 24 million years later[18].

## Methods

This research complies with all relevant ethical regulations. The fossil excavation was permitted by Department of Natural Resources and Department of Science & Technology in Tibetan Autonomous Region, China.

**Geological setting**. Fossils in this study are from the Dayu and Xiede sections, which are located within the Bangong-Nujiang Suture Zone that is aligned roughly East-West through what is now the central Tibetan Plateau. The present average surface elevation of these two fossil sites is about 4700 metres[42], and predominantly vegetated by alpine steppe[34] (Supplementary Figure 1).

The Bangong-Nujiang Suture Zone hosts thick Cenozoic deposits, the lower part of which in the adjacent Lunpola and Bangor basins comprise the Paleocene-Eocene Niubao Formation and the upper part the Oligocene-Miocene Dingqinghu (Dingqing) Formation[75-77]. The Niubao Formation, up to 3,000 metres thick[36], is mainly formed of fluviatile red clastic rocks interspersed with greyish green mudstones and sandstones. Plant fossils have recently been reported from mudstones in the middle part of the formation near the top of the lower member of the Niubao Formation, and include, amongst others, *Ailanthus maximus*[49], *Lagokarpos tibetensis*[78], *Illigera eocenica*[79], and *Asclepiadospermum marginatum* and *A. ellipticum*[80].

The Dingqinghu Formation is characterised by grey lacustrine mudstones together with sandstones and oil shales, with a total thickness of about ~1,000 metres[36,42,76]. It bears rich records of animals and plants, including two Oligocene ostracode assemblages: *Austrocypis-Cyprinotus-Pelocypris* and *Ilyocypris-Limnocythere*[81,82], fishes including *Plesioschizothorax*[83], and mammals such as *Plesiaceratherium*[42].

Fossils in this study are from two sites in the Niubao Formation (Fig. 1). According to recent radiometric dating throughout the Dayu section, the age of the plant fossil horizons in the Dayu section (32° 20′ N, 89° 46′ E) is ~39 Ma[36] (Bartonian, early late Eocene). For the other site in Xiede (31° 58′ N, 88° 25′ E), the biota is similar to that at Dayu, and many species are common to both sites, such as a climbing perch *Eoanabas thibetana*[84], a water strider *Aquarius lunpolaensis*[85], and plants, e.g., *Limnobiophyllum pedunculatum*[86], *Sabalites tibetensis*[51], *Cedrelospermum tibeticum*[50], and *Ailanthus maximus*[49]. Therefore, we consider both sites to be age-equivalent, dating from the early late Eocene.

**Morphological observations**. In total, 44 spiny fossils were collected from the Dayu and Xiede localities. Among them, 19 specimens are from the Dayu section, and 25 specimens are from the Xiede section, respectively. All specimens are deposited at the Palaeoecology Collections of Xishuangbanna Tropical Botanical Garden (XTBG), Chinese Academy of Sciences. They were photographed with a Nikon D700 digital camera (Nikon, Kanagawa, Japan). A stereoscope (Zeiss Smart Zoom 5) was used to photograph the detailed morphology of the spines at the

Central Laboratory in XTBG. Digital photographs of extant and fossil specimens were measured using ImageJ 1.52a software (http://rsb.info.nih.gov.ig/).

The morphological definitions of spines followed Cornelissen[2], and Bell and Bryan[87]. We compared spine morphologies in living eudicots and monocots (Supplementary Fig. 15) to those of our fossils and found that only eudicots exhibit the same spine morphology[87] as those of our fossils. We classified fossil spines into prickles and thorns according to two criteria: prickles originate from the epidermis of plant organs such as stems, leaves and petioles, whereas thorns are modified branches and include internal vascular bundles. For fossil specimens, due to the different structures of prickles and thorns, we observed distinct scars at the junction between prickles and stems, but these are absent in thorny species. Secondly, thorns always grow from stem nodes and therefore display a regular phyllotaxy, but prickles tend to be distributed randomly along the internodes.

For morphological comparison of spinescence among modern species, we consulted online data sources including JSTOR Global Plants (http://plants.jstor.org/) and the Chinese Virtual Herbarium (http://www.cvh.org.cn/). Living species of *Rubus alceifolius* and *Xylosma racemosum* were collected from XTBG and from the wild in the Xishuangbanna region, Yunnan.

**Phylogenetic analyses**. To investigate the emergence and early diversification of spiny eudicots, we reconstructed the evolutionary history of spines across species of woody eudicots represented on the mega-phylogeny of plants from Zanne et al.[88] using the make.simmap function in phytools[89] and the ape[90] and Geiger[91] R-libraries (Supplementary Code 1). We coded, as a binary trait, for the presence of spines in plants found in Eurasia (n = 1590 species) mostly using scans from https://plants.jstor.org/. We excluded species when the origin appeared ambiguous between more than one continent. To control for the relatively recent origin and diversification of spines, we allowed rates to vary through time (parameter Δ = 9), and constrained evolutionary transitions to preclude reversals from the spiny to non-spiny state, reflecting the rarity of this evolutionary event, and avoiding potential bias due to the over-representation of non-spiny lineages among woody taxa[92].

**Palaeoenvironmental reconstruction using Climate-Leaf Analysis Multivariate Program (CLAMP)**. We applied CLAMP (http://clamp.ibcas.ac.cn) to the woody dicot leaf forms (24 distinct morphotypes in the Dayu flora, Supplementary Figures 9, 10) with the PhysgAsia2 (http://clamp.ibcas.ac.cn/CLAMP_PhysgAsia2.html) training set and Worldclim2 climate data (http://worldclim.com/version2) calibration[74] to derive palaeoclimate metrics. Scoring of the fossil leaves followed the CLAMP protocols. The CLAMP scoresheets are given as Supplementary Tables 2, 3.

Among the palaeoclimate metrics that CLAMP returns in this calibration are moist enthalpy and wet bulb mean annual temperature and both can be used to assess palaeoelevation. The former uses conservation of energy principles, and the latter exploits the reduction in temperature that occurs as a land surface increases in height (a terrestrial thermal lapse rate). Both approaches were used to determine the height at which the Dayu flora existed.

As a parcel of air rises against a mountain front, temperature declines, humidity rises and potential energy increases, but overall the energy it contains is conserved[93]. This energy is called moist static energy (h) and excludes kinetic energy, which tends to be small except during a hurricane.

$$h = H + Zg \qquad (1)$$

where H is moist enthalpy, Z is height, and g is gravitational acceleration (a constant 9.81 cm/s²).

This conservation of energy means that we can exploit the difference in moist enthalpy between two locations ($H_{low}$ and $H_{high}$) to determine the height difference between them, $\Delta Z$, as follows:

$$\Delta Z = (H_{low} - H_{high})/g \qquad (2)$$

Thermal lapse rates describe changes in temperature with changing elevation, and in general (with exception of a temperature inversion) temperature measured at Earth's surface declines with increasing height and can be expressed as:

$$\Gamma\varepsilon = -dT/dz \qquad (3)$$

where $\Gamma\varepsilon$ is the terrestrial lapse rate, $dT$ is the change in temperature, and $dz$ is the change in height. We used wet bulb lapse rates because these are more reliable than those based on dry bulb temperatures and are independent of season[38]. The local terrestrial wet bulb lapse rate was determined from modelling as in Farnsworth et al.[38] using a best estimate realistic topography.

Sea level moist enthalpy was derived from model data adjusted to be compatible with CLAMP values using Eocene archived proxy data ranging from northern India to Svalbard, adjusted for palaeolatitude, following the methodology of Su et al.[33]. Wet bulb temperature at mean sea level was also obtained from the adjusted model data. Further details are given in Supplementary Note 2.

**Palaeovegetation predicted by climate modelling**. To produce a more applicable, time dependant, simulation of the Dayu environment, we employed a fully coupled Atmosphere-Ocean General Circulation Model (AOGCM), HadCM3BL-

M2.1aD[94]. Model boundary conditions (topography, bathymetry, and ice sheet configurations; at 0.5×0.5° resolution and downscaled to model resolution) for the Bartonian (~39 Ma) are provided by Getech Plc. Stage-specific solar luminosity was calculated using the methods of Gough[95]. Atmospheric $CO_2$ concentrations were prescribed at 1,120 ppm, consistent with the Phanerozoic $CO_2$ compilation of Foster et al.[96]. Each experiment was run for 12,422 model years to allow the surface and deep ocean to reach equilibrium and achieve a state with no net energy imbalance at the top of the atmosphere. This is fundamental as ocean circulation can take many thousands of model years to establish its equilibrium state. Climate means were calculated from the last 100-years of each simulation. Time-varying latitude and longitude plate palaeo-rotations were provided for the Dayu location to allow for accurate comparison within the model. Dayu was located at a palaeolatitude of 32° 30′ N, 82° 54′ E using the Getech Plc. plate model. Tibetan orography was constrained to represent high (5 km) Gangdese and Tanggula mountain systems bounding an East-West trending valley system the floor of which was set at 2.5 km (Supplementary Fig. 14A). The dynamic vegetation scheme TRIFFID[97,98] (Top-down Representation of Interactive Foliage and Flora Including Dynamics), which predicts the distribution and properties of global vegetation based on plant functional types using a competitive, hierarchical formulation derived from the land-atmosphere climate interactions in the model (TRIFFID was executed and updated every 10 model days) to predict vegetation types likely to existed within the valley (Supplementary Fig. 14B).

**Herbaceous plant and vegetation types in the late Eocene**. To understand the vegetation type, we assessed the diversity of grass species in the flora. We studied megafossils, pollen grains/spores, and phytoliths indicative of grass from the spiny plant fossil-bearing layer of the Dayu section. Pollen grains and spores were poorly preserved in Dayu and Xiede sections, preventing us from further palynological investigation, but phytoliths were relatively well preserved. The extraction and identification of phytoliths from sediments followed the methods of Strömberg[99] and Lu[100]. The procedure consisted of treatment with 30% hydrogen peroxide ($H_2O_2$) and cold 15% hydrochloric acid (HCl), followed by heavy liquid separation using zinc bromide ($ZnBr_2$, density 2.35 g/cm$^3$) and mounting on a microscope slide with Canada balsam. From each extracted sample, at least one slide was prepared for phytolith counts and analysis under a LEICA DM 1000 microscope at 400× magnification. For these samples with poorly preserved phytoliths somewhat less than 100 diagnostic forms were counted (Supplementary Table 4).

**Paleogene herbivorous mammal fossils**. We used fossil records of all Artio-dactyls and Perissodactyls from fossilworks (http://fossilworks.org/) (Supplementary Data 1). Both are orders including mammals that have similar ecological roles as modern ungulates. From fossil records, we built a dataset with the first occurrence of each species using the maximum date estimate as the species are very likely older than the first fossil. This allowed us to deduce a cumulative curve of species ($n = 658$ species) as an estimate of speciation rate. We focused on the fossil records from countries including and surrounding the Tibetan Plateau, including China, India, Afghanistan, Kazakhstan, Kyrgyzstan, Mongolia, Myanmar, Nepal, Pakistan, Turkmenistan, and Uzbekistan.

**Reporting summary**. Further information on research design is available in the Nature Research Reporting Summary linked to this article.

## Data availability

All data analysed in this paper are available as part of the Article and Supplementary Information. All fossils are deposited at the Paleoecology Collections of Xishuangbanna Tropical Botanical Garden (XTBG), Chinese Academy of Sciences. Additional data related to this paper may be requested from the authors. Correspondence and requests for materials should be addressed to T.S. (sutao@xtbg.org.cn). Source data are provided with this paper.

## Code availability

Code for phylogeny reconstruction is provided in the Supplementary Information.

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

## Acknowledgements

We are grateful to members of the Paleoecology Research Group at Xishuangbanna Tropical Botanical Garden (XTBG) and Institute of Vertebrate Paleontology and Paleoanthropology (IVPP) who participated in numerous fossil collection expeditions on the Tibetan Plateau. We thank Professor Houyuan Lu and Professor Caroline A.E. Strömberg for contributions to the identification of phytoliths; the Central Laboratory of Public Technology Service Center of XTBG for help with photography. This work is supported by National Natural Science Foundation of China (NSFC) (Nos. 41988101 and 41922010), the Second Tibetan Plateau Scientific Expedition programme (No. 2019QZKK0705), Natural Environment Research Council of the UK (NERC) (Nos. 41661134049 and NE/P013805/1), the Strategic Priority Research Program of the Chinese Academy of Sciences (CAS) (Nos. XDA20070301 and XDB26000000), Youth Innovation Promotion Association, CAS (No. Y2021105), and the West Light Foundation, CAS (No. 2020000023).

## Author contributions

T.S and Z.-K.Z. designed research. X.-W.Z., U.G., R.A.S., K.W.T., and T.S. assembled the data for the manuscript and led the writing process. X.-W.Z., U.G., F.-X.W., P.-R.C., C.D.R., J.L., J.H., X.-T.X., S.-T.Z., K.W.T., T.D., Z.-K.Z. and T.S. collected fossil data. U.G. coded the presence of spine in the phylogeny and conducted the trait ancestral reconstruction. R.A.S., A.F., S.-F.L., and P.J.V. performed the numerical climate and elevation model analyses. X.-W.Z. and T.S. wrote the first draft of the paper. X.-W.Z., U.G., R.A.S., K.W.T., and T.S. revised the manuscript. All authors discussed and commented on the manuscript.

## Competing interests

The authors declare no competing interests.
