## [Peer Review File · Nature Communications]

Rapid Eocene diversification of spiny plants in subtropical woodlands of central TibetReviewers' Comments:

Reviewer #1:

Remarks to the Author:

This research presents a remarkable association of late Eocene plants that includes macrofossils and phytoliths from central Tibet. The study is accompanied by phylogenetic analysis of spiny eudicot plants, numerical climate (CLAMP) and vegetation models, and also accumulation curve of mammalian herbivore species in central Asia. I applaud that the authors collected this new fascinating fossil material from the Niubao Formation. I agree that the relative abundance of late Eocene spiny plants in central Tibet will be essential to understanding plant defense mechanisms' evolution. These types of spiny fossil plants are indeed rare in the recent geological record. Much of the writing is clear, but some aspects of the manuscript left me wondering:

1. Is there really a rapid Eocene diversification of spiny plants in subtropical woodlands of central Tibet? This is perhaps my major concern of the paper. To make this case brief, just by looking at fig 3a, the red line representing the proportion of spiny plants in eudicots in Eurasia during the Cenozoic only increases "rapidly" around 20 million years (Miocene time). From ~39 Ma (the age of the new fossils) to ~20 Ma, there is almost a steady gap in the spiny species accumulation curve, which contrasts highly with the mammalian herbivore curve. I see an early origination of spiny plants based on the fossils reported here, but the "rapid diversification" is not convincing.
2. All spiny plants were described as eudicots – I think this is a reasonable assumption considering the fragmentary nature of the specimens. However, palm stems and other monocots are commonly spiny too. Palm fossils are also present in the Dayu section. How can the authors discard a monocot affinity?
3. Regarding point number two and the overall composition of the flora (supplements 8-10). None of those fossil taxa and families described are typically spiny in my opinion. They look very similar to other Eurasian floras. I think that the "open woodland landscape" reconstruction needs further investigation.

Last, I believe this paper could provide an excellent opportunity to understand the evolution of a rare type of plant mechanisms seen in the fossil record. I hope the authors can improve the manuscript.

Reviewer #2:

Remarks to the Author:

This is an excellent manuscript, describing a very interesting assemblage of spiny plants. It is very exiting to see such a detailed assessment of a spiny flora, especially when linked to patterns of mammal herbivory; spines are so common and important in modern plants, and understudied in fossil plants.

Comments:

Can you expand at all on how the function of spinescence varies between prickles and thorns, or for the different prickle and thorn morphologies? Do they defend against different sizes of herbivores, or against different feeding behaviors?

Are there any other defensive structures (than spines and phytoliths) in the flora? For example any spiny edges or trichomes on the leaves?

line 167: change 'amphibian' to 'amphibious'

Reviewer #3:

Remarks to the Author:

Dear Editor Nature Communications:

Thank you so much for inviting me as a reviewer of the manuscript entitled: "Rapid Eocene diversification of spiny plants in subtropical woodlands of central Tibet" for Nature Communications". I have reviewed this manuscript according to your suggestions. Please find my comments attached to this letter.

In this study, the authors provided Macro spiny plant fossils and a few phytoliths in the same deposits in Dayu and Xiede sections around ~39 Ma ago. The results suggest that climate aridification and expansion of herbivorous mammals, have significant impact on the diversification of the spiny plants. I think this work will make a valuable contribution to understand the relationship among vegetation and herbivorous mammals' evolution, and climate change on the Tibetan Plateau.

However, I have a few major concerns. These important questions should be address clearly.

There are abundant and various phytolith types in herbaceous plants, especially Poaceae. On the contrary, phytolith in arboreal and shrubby plants are relatively low in abundance. In addition, calciphilous plants don't have phytolith. The limited phytolith plates provide the incomplete vegetation assemblage. Thus, single microfossil proxy, phytolith, is not reliable to reveal the grassland vegetation assemblage. Furthermore, phytolith assemblage is not shown in this manuscript. I can't judge the herbaceous vegetation condition, either.

Pollen proxies may be much more effective than phytolith and can provide more widely information than phytolith which may provide vegetation dynamic among arboreal, shrubby, and herbaceous plants. I suggest providing pollen and phytolith assemblage results in revised manuscript.

Moreover, phytolith nomenclatures contains a few mistakes. Sup Figure 12 B should not be identified as rondel due to the saddle top, it is much more likely a saddle, notice the rondel in SF 12 D, it has a flat top. Figure 12E is not a typical bamboo fan, just identify it as a common fan is ok, so it is the same with SF 12F, the common fan could be found in several subfamilies in Poaceae. Figure 12H is highly doubted to be a phytolith. The phytolith results (Fig. 4 and S12) were not convincing, and please refer to the ICPN 2.0 for the nomenclature of phytolith types. (Taxonomy, I.C.f.P., 2019. International Code for Phytolith Nomenclature (ICPN) 2.0. *Annals of Botany* 124, 189-199; MADELLA, M., ALEXANDRE, A., BALL, T., GROUP, I.W., 2005. International Code for Phytolith Nomenclature 1.0. *Annals of Botany* 96, 253-260.)

In figure 1, spiny plant fossils were found in four points, such as DY1, DY2, XDA1 and XDB3. Are spiny plants still found at the later sequence of the two profiles? If not, Why? The vegetation evolution of microfossils evidence before and after ~39Ma should be provided, confirming that the process of aridification triggering the increases of spiny plants.

Detailed comments:

Line 12 & 62

Phytolith fossil evidence is insufficient to prove the open woodland landscape.

Line 184

How do the authors come to "which may be because spinescence tends to be most prevalent in semi-arid to arid environments".

Line 199

If there are abundant phytoliths, please show the phytolith assemblages with a figure or table.

Line 201

The common bulliform (fan-shaped) is not only belong to Bambusoideae but also belong to Oryzoideae, Chloridoideae, Arundinoideae, Panicoideae. No bamboo diagnostic phytolith has been provided the authors. Nowadays, Bambusoideae indicates monsoon dominant climate, which is widely distributed in East Asian monsoon region. So it is questionable to announce open ecosystem based on bulliform phytolith.

Due to the insufficient microfossil proxy, this manuscript does not meet the standard of this high impact factor journal.

Response to Reviewer #1

This research presents a remarkable association of late Eocene plants that includes macrofossils and phytoliths from central Tibet. The study is accompanied by phylogenetic analysis of spiny eudicot plants, numerical climate (CLAMP) and vegetation models, and also accumulation curve of mammalian herbivore species in central Asia. I applaud that the authors collected this new fascinating fossil material from the Niubao Formation. I agree that the relative abundance of late Eocene spiny plants in central Tibet will be essential to understanding plant defense mechanisms' evolution. These types of spiny fossil plants are indeed rare in the recent geological record. Much of the writing is clear, but some aspects of the manuscript left me wondering:

Response: We appreciate these very positive comments and constructive suggestions, and we have revised the manuscript by following all suggestions proposed. Please see our point-by-point responses as below.

1. Is there really a rapid Eocene diversification of spiny plants in subtropical woodlands of central Tibet? This is perhaps my major concern of the paper. To make this case brief, just by looking at fig 3a, the red line representing the proportion of spiny plants in eudicots in Eurasia during the Cenozoic only increases "rapidly" around 20 million years (Miocene time). From ~39 Ma (the age of the new fossils) to ~20 Ma, there is almost a steady gap in the spiny species accumulation curve, which contrasts highly with the mammalian herbivore curve. I see an early origination of spiny plants based on the fossils reported here, but the "rapid diversification" is not convincing.

Response: We agree that our original figure did not appear to clearly show the exponential increase of spiny plants. In order to make our point better, we present a new figure with a log plot instead (Figure 3a). In such a plot an exponential increase translates to a straight line, and we have now highlighted this in the figure legend. This type of plot was used in Charles-Dominique et al., (2016, PNAS, E5572-E5579; cited in the revision), showing the rapid speciation more clearly. We also clarify this in the text (lines 110-122 in the revision). According to our original data, only 2 spiny lineages emerged in the first 24 Myr, but in the next 10 Myr, there was a four-fold increase from 2 to 8 spiny lineages matching well the increase of 1 to 7 morphotypes for our fossils. We would also expect a delay in the response of plants to herbivory because not all herbivore clades (elephant, hyrax) are deterred by spines as suggested by Charles-Dominique et al., (2016). We believe the modifications we have made in response to reviewer comments will make the rapid diversification of spiny plants more apparent.

2. All spiny plants were described as eudicots – I think this is a reasonable assumption considering the fragmentary nature of the specimens. However, palm stems and other monocots are commonly spiny too. Palm fossils are also

present in the Dayu section. How can the authors discard a monocot affinity?

Response: Yes, there are some monocots, such as palms, that do have spines, but they can be distinguished from the spines of dicots by their morphology and that they only occur on the characteristic palm petiole. The spine specimens we recovered were found intact on clearly woody stems. Those stems have the characteristic of eudicots, not monocots. Of the spines that were found unconnected to stems, all appear to be prickles. Because prickles on eudicots arise from epidermal or subepidermal layers and lack vasculature, they easily detach from stems (e.g., roses). Palm spines occur on the rachis of palm leaves, and they may look like prickles (Supplementary Figure 15), but they are much more firmly attached to the rachis than eudicot prickles; so, we consider it most unlikely that we would find palm spines as detached objects. We have added a plate (Supplementary Figure 15) to show the difference. Moreover, spiny structures are not observed in our palm fossils from the same layer (Figure 2 in Su et al., 2019, Science Advances, eaav2189). We referred to the book 'Plant Form' (Part 1 in Bell, A. D. & Bryan, A., 2008) for a detailed description on the classification and distribution of spines in living plants. Only eudicots, but no monocots, present the same spines as seen in our fossils. We have added a sentence to make the statement more clearly (Lines 380-382 in the revision):

SENTENCES ADDED: We compare spine morphologies in living eudicots and monocots (Supplementary Figure 15) to those of our fossils and find that only eudicots exhibit the same spine morphology as those of our fossils.

3. Regarding point number two and the overall composition of the flora (supplements 8-10). None of those fossil taxa and families described are typically spiny in my opinion. They look very similar to other Eurasian floras. I think that the "open woodland landscape" reconstruction needs further investigation.

Response: Our census of the leaf flora from the same layer in the Dayu site shows that the modern affinities of 10 fossil taxa in the flora potentially have spines, i.e., Malvaceae, Rosales, Fabaceae, Ulmaceae, Cannabaceae, Menispermaceae, Simaroubaceae, Anacardiaceae, Myrtaceae, Araliaceae. We have added this information in Supplementary Table 6 in the revision. We have also added a sentence in the revision to demonstrate that (Lines 257-261 in the revision):

SENTENCES ADDED: Our census of the leaf flora from the same layer at the Dayu site shows that the modern affinities of 10 fossil taxa in the flora belong to families/orders that contain spine-bearing species, i.e., Malvaceae, Rosales, Fabaceae, Ulmaceae, Cannabaceae, Menispermaceae, Simaroubaceae, Anacardiaceae, Myrtaceae, and Araliaceae (Supplementary Table 6).

Last, I believe this paper could provide an excellent opportunity to understand the evolution of a rare type of plant mechanisms seen in the fossil record. I

hope the authors can improve the manuscript.

Response: Thank you again, we believe this submission has been improved substantially by following your suggestions.

Response to Reviewer #2

This is an excellent manuscript, describing a very interesting assemblage of spiny plants. It is very exciting to see such a detailed assessment of a spiny flora, especially when linked to patterns of mammal herbivory; spines are so common and important in modern plants and understudied in fossil plants.

Response: Thank you for emphasizing the importance of our work. We have revised the manuscript following all your suggestions.

Comments:

Can you expand at all on how the function of spinescence varies between prickles and thorns, or for the different prickle and thorn morphologies? Do they defend against different sizes of herbivores, or against different feeding behaviors?

Response: This is a good question, but this is not possible to answer with our current knowledge because so far, there is no evidence that thorns and prickles have different functions in term of overall plant defence. Studies have shown that both prickles and thorns can decrease herbivory (Cooper and Owen-Smith, 1986; Milewski et al., 1991) and even prevent climbing (Cooper and Ginnett, 1998). However, the specific link between spine shape and their function still needs to be explored as spines can both affect directly the biomass removed, but also, they may have more indirect effects by influencing herbivory behaviour in different ways. So, unfortunately, we are not able to answer these questions with the current knowledge, but future research is aiming at testing these questions. For example, thorns can form defence mesh that can only be penetrated by certain types of feeding behaviour, while prickles present a more direct feeding deterrent. However, production of prickles versus thorns might be determined simply by plant developmental constraints; for some families, it is easier to generate spine from one tissue and for others from another one, which would result in different defence architectures. Please check lines 37-39 in the revision.

SENTENCES ADDED: Both prickles and thorns can defend against herbivores and even climbing mammals, but whether they have other functional properties is unclear.

Are there any other defensive structures (than spines and phytoliths) in the flora? For example, any spiny edges or trichomes on the leaves?

Response: We did not find any spiny edges or trichomes on the leaves. In these deposits, the trichomes were not preserved because the leaf fossils are just impressions. Moreover, trichomes could also be produced to increase the

leaf boundary layer as thus reduce water loss and may not have a primary defence role against mammalian herbivory.

line 167: change 'amphibian' to 'amphibious'

Response: Yes, we have changed 'amphibian' to 'amphibious'. Please check line 205 in the revision.

Response to Reviewer #3

In this study, the authors provided Macro spiny plant fossils and a few phytoliths in the same deposits in Dayu and Xiede sections around ~39 Ma ago. The results suggest that climate aridification and expansion of herbivorous mammals, have significant impact on the diversification of the spiny plants. I think this work will make a valuable contribution to understand the relationship among vegetation and herbivorous mammals' evolution, and climate change on the Tibetan Plateau.

However, I have a few major concerns. These important questions should be address clearly.

Response: We thank the reviewer #3 for the comments. We have revised the manuscript following all these suggestions. Please check our point-by-point responses as below.

There are abundant and various phytolith types in herbaceous plants, especially Poaceae. On the contrary, phytolith in arboreal and shrubby plants are relatively low in abundance. In addition, calciphilous plants don't have phytolith. The limited phytolith plates provide the incomplete vegetation assemblage. Thus, single microfossil proxy, phytolith, is not reliable to reveal the grassland vegetation assemblage. Furthermore, phytolith assemblage is not shown in this manuscript. I can't judge the herbaceous vegetation condition, either.

Response: We agree with the reviewer that the limited phytoliths shown on the plates could represent a partial vegetation assemblage. In the revision, we have re-examined the phytoliths type and abundance throughout the Dayu section and added figures and tables to show all phytolith types (Supplementary Figure 12, Supplementary Table 4). The results indicate that spiny fossil-bearing layer of the Dayu section preserves abundant phytoliths, with 63% being those of grasses and 37% of woody plants among the identifiable types. By contrast, in the lower parts of the Dayu section, phytoliths are dominated by those produced by woody plants, and the phytoliths of grasses are mostly bulliform, which is a common type in grasses. We also examined the phytolith composition of the older nearby Jianglang flora reported by Su et al., (2020) and in that assemblage there are very few grass phytoliths, as would be expected in the more closed subtropical forest reconstructed from the megafossils. It seems that phytolith type and

abundance track developmental changes in vegetation through time as reconstructed by both the megafossils and climate/vegetation modelling. We have revised the sentences accordingly (Lines 157-164 in the revision).

Furthermore, in this study, we are not relying solely on phytolith diversity to describe the overall nature of the palaeovegetation. We have used other proxies, together with phytoliths, to reveal the structure and composition of the vegetation, in this case woodland. We include herbaceous and non-herbaceous megafossils as well as model simulations. The abundant phytoliths demonstrate a rich herbaceous component in the vegetation at that time, which would otherwise not be recorded in herbaceous plant material with low preservation potential. It is probably true to say that phytoliths are not normally looked for and recorded when other plant material such as leaf, fruit and seed megafossils are evident, so this study is exceptional in that respect. The phytoliths we describe are in addition to the other fossil evidence that used for determining the palaeovegetation. Nevertheless, as suggested by the reviewer, we have added more information on the phytolith assemblage of the Dayu section in the revision (Lines 172-175 in the revision; Supplementary Figure 12). We have also added the phytolith assemblage from the Jianglang section (subtropical forest, ~47 Ma; Su et al., 2020, PNAS, 32989-32995) near the Dayu site for comparison (Supplementary Figure 13), which shows a high phytolith composition of woody plants. We have added a sentence to show that (Lines 165-175 in the revision).

SENTENCE ADDED: Progression from the older flora to younger in the Bangor and Lunpola basins shows an increase in phytolith abundance and diversity. The phytolith assemblage from the ~47Ma Jianglang section, where the megafossils are interpreted to represent a subtropical forest, is dominated by those forms produced by woody taxa, and these greatly exceed grass forms (Supplementary Figure 13, Supplementary Table 4). Within the lower part of the Dayu section (~39 Ma), phytoliths are still dominated by those produced by woody plants but bulliform phytoliths typically produced by grasses become more common (Supplementary Figure 13). The main fossil-bearing layer higher in the Dayu section preserves not only the numerous spiny taxa but also abundant phytoliths where, among those that could be identified, 63% were produced by grasses and only 37% by woody plants.

Pollen proxies may be much more effective than phytolith and can provide more widely information than phytolith which may provide vegetation dynamic among arboreal, shrubby, and herbaceous plants. I suggest providing pollen and phytolith assemblage results in revised manuscript.

Response: Yes, pollen proxies can be effective to show the vegetation dynamics, notwithstanding their long-distance transport and mixing. We have carried out palynological analysis in Dayu and Xiede sections, unfortunately, there are only few pollen grains and spores and those that do occur are poorly preserved, preventing us from further palynological investigation. We have

added a sentence in the revision to clarify that (Lines 476-478 in the revision):
SENTENCES ADDED: Pollen grains and spores were poorly preserved in Dayu and Xiede sections, preventing us from further palynological investigation, but phytoliths were relatively well preserved.

Moreover, phytolith nomenclatures contains a few mistakes. Sup Figure 12 B should not be identified as rondel due to the saddle top, it is much more likely a saddle, notice the rondel in SF 12 D, it has a flat top. Figure 12E is not a typical bamboo fan, just identify it as a common fan is ok, so it is the same with SF 12F, the common fan could be found in several subfamilies in Poaceae. Figure 12H is highly doubted to be a phytolith. The phytolith results (Fig. 4 and S12) were not convincing, and please refer to the ICPN 2.0 for the nomenclature of phytolith types. (Taxonomy, I.C.f.P., 2019. International Code for Phytolith Nomenclature (ICPN) 2.0. *Annals of Botany* 124, 189-199; MADELLA, M., ALEXANDRE, A., BALL, T., GROUP, I.W., 2005. International Code for Phytolith Nomenclature 1.0. *Annals of Botany* 96, 253-260.)

Response: We have carefully re-examined all the phytoliths using the International Code for Phytolith Nomenclature and double checked the identification. Moreover, we have consulted with Professor Houyuan Lu and Professor Caroline A.E. Strömberg who are experts in the field in regard to the identification of phytoliths. We have acknowledged their contribution in the Acknowledgements. Please check Supplementary Figure 12 in the revision.

In figure 1, spiny plant fossils were found in four points, such as DY1, DY2, XDA1 and XDB3. Are spiny plants still found at the later sequence of the two profiles? If not, Why? The vegetation evolution of microfossils evidence before and after ~39Ma should be provided, confirming that the process of aridification triggering the increases of spiny plants.

Response: The preservation of fossils in the central part of the Tibetan Plateau is spatially very variable. They tend to be preserved only in specific lacustrine successions and the greatest density of fossils only occur in facies close to the ancient shoreline. Limited surface exposure of deformed Eocene strata, and the reporting only of ones that are well (radiometrically) dated, means useful fossil-bearing outcrops are hard to locate. The few layers we report are the result of 6 years of intensive fieldwork at almost 5,000m altitude. Regarding the process of ongoing aridification and environmental change we can now refer to a recent study by Xiong et al. (2022) that documents facies climate and elevation change through the Niubao and Dingqqinghu Formations in a new chronostratigraphic framework based on radiometric dating and seismic profiles. This work demonstrates both drying and cooling between 39 and 29 Ma as surface elevation increased from < 2.5 km to > 4 km. Megafossils disappeared from the succession as mean annual temperature fell to near freezing (~1 °C) by 29 Ma (early Oligocene), the later Oligocene warming through to the mid Miocene reintroduced temperate woodlands as

indicated by younger pollen assemblages in the Dingqinghu formation, which overlies the Niubao in the BNSZ basins. We have cited this recent research (Xiong et al., 2022) and revised the sentences accordingly (Lines 303-305 in the revision).

Detailed comments:

Line 12 & 62

Phytolith fossil evidence is insufficient to prove the open woodland landscape.

Response: We agree that phytolith fossil evidence is insufficient alone to demonstrate an open woodland landscape. As mentioned above, we have added detailed information on phytolith assemblages in the Dayu section and compared them with the phytolith assemblage in an older flora (~47 Ma) presenting by subtropical forest (Lines 165-175 in the revision; Supplementary Figure 13, Supplementary Table 4). The Dayu woodland vegetation type is further supported by other evidence including numerous herbaceous megafossils, some of which are aquatic, but most are terrestrial. This diverse presence of Poaceae is complemented by a variety of woody dicot fossils which evidence trees and shrubs. Our climate and vegetation modelling, as well as the rich herbivory mammal fossils also support our interpretation. We have rephrased the sentences in the revision (Lines 13-15 and 67-69).

Line 184

How do the authors come to “which may be because spinescence tends to be most prevalent in semi-arid to arid environments”.

Response: To support this contention we have added a reference (Charles-Dominique et al., 2016, PNAS E5572-E5579) to show that spinescence tends to be most prevalent in semi-arid to arid environments in modern ecosystems. Please check line 229 in the revision.

Line 199

If there are abundant phytoliths, please show the phytolith assemblages with a figure or table.

Response: We have added a figure to show the complete phytolith assemblages in Dayu section. Please check Supplementary Figure 12 in the revision.

Line 201

The common bulliform (fan-shaped) is not only belong to Bambusoideae but also belong to Oryzoideae, Chloridoideae, Arundinoideae, Panicoideae. No bamboo diagnostic phytolith has been provided the authors. Nowadays, Bambusoideae indicates monsoon dominant climate, which is widely distributed in East Asian monsoon region. So, it is questionable to announce open ecosystem based on bulliform phytolith.

Due to the insufficient microfossil proxy, this manuscript does not meet the

standard of this high impact factor journal.

Response: We have checked the identification of phytoliths again (including the fan type) as well as consulting colleagues (Professor Houyuan Lu and Professor Caroline A.E. Strömberg) regarding the identification of phytoliths. We agree that bulliform phytoliths are not unique to bamboos, they may belong to other groups such as Oryzoideae, Chloridoideae, Arundinoideae, and Panicoideae. The range of other bulliform producers mentioned by the reviewer (Oryzoideae, Chloridoideae, Arundinoideae, Panicoideae) just strengthen our interpretation that a variety of grasses were present in the Dayu ecosystem. The main purpose of using phytoliths is to provide evidence on the diversity of the assemblage of grass species in the ecosystem to complement other fossil evidence for the tree/shrub components of the vegetation. We have added a sentence to clarify this (Lines 160-164 in the revision):

SENTENCES ADDED: The diverse morphology of phytoliths observed from the fossil-bearing outcrops (Supplementary Figure 12) indicates a species-rich grass community within an ecosystem that also contained palms and numerous other woody taxa, but evidence for such a diverse grass component does not occur in older Tibetan floras, e.g., the middle Eocene Jianglang flora from central Tibet (Supplementary Figure 13).

Reviewers' Comments:

Reviewer #2:

Remarks to the Author:

All of my concerns have been addressed in the revision, and I have no further comments.

Reviewer #3:

Remarks to the Author:

Significant improvements and corrections have been made to this version of the manuscript. The authors replied to questions quite well. The manuscript almost appears to be up to the standards of this journal.

However, I still have some concerns about the phytolith identification plate in SI Figure 12 after communications with Prof. Houyuan Lu and other phytolith researchers.

1. There are two options for J: One is elongate shaped; the other is point shaped. Thus, I suggest removing this ambiguous phytolith photo in the final version.

2. M & N may come from the motor cell of Poaceae plants, which don't belong to woody plants.

3. Please recalculate percentages of woody and grass plants, after re-examining the phytolith identification data.

These errors need to be carefully corrected before publication.

Response to Reviewer #2

All of my concerns have been addressed in the revision, and I have no further comments.

Response: Thank you for all your constructive suggestions in the first revision.

Response to Reviewer #3

Significant improvements and corrections have been made to this version of the manuscript. The authors replied to questions quite well. The manuscript almost appears to be up to the standards of this journal.

However, I still have some concerns about the phytolith identification plate in SI Figure 12 after communications with Prof. Houyuan Lu and other phytolith researchers.

1. There are two options for J: One is elongate shaped; the other is point shaped. Thus, I suggest removing this ambiguous phytolith photo in the final version.

Response: Thank you. We have removed the ambiguous phytolith photo J in Supplementary Figure 12. Please see Supplementary Figure 12 in the revision.

2. M & N may come from the motor cell of Poaceae plants, which don't belong to woody plants.

Response: We agree and we have assigned M to bulliform shaped belonging to Poaceae; meanwhile, we have assigned N to the undetermined type because of its poor preservation. Please see the legend of the reorganized Supplementary Figure 12.

3. Please recalculate percentages of woody and grass plants, after re-examining the phytolith identification data. These errors need to be carefully corrected before publication.

Response: We have recalculated percentages of woody and grass plants after re-examining the phytolith identification data. Only the DY-Fossil-bearing layer presents the phytolith types mentioned above; therefore, we recalculated the percentages of woody and grass plants in this layer in the revision. According to the recalculation, 66% (previously 63%) phytoliths were produced by grasses and only 34% (previously 37%) by woody plants; therefore, this recalculation does not change our main conclusion on the existence of semi-open habitat dominated by herbaceous plants in central Tibet during the late Eocene. Please check Lines 173-174 in the main text, Supplementary Figure 13, and Supplementary Table 4 in the revision.

Reviewers' Comments:

Reviewer #3:

Remarks to the Author:

Well done! It's all clear. I have no more comments.

Response to Reviewer #3

Well done! It's all clear. I have no more comments.

Response: Thank you again for all your constructive suggestions.